# Key factors influencing motivation among health extension workers and health care professionals in four regions of Ethiopia: A cross-sectional study

**Mehiret Abate**[1]*, **Zewdie Mulissa**[1], **Hema Magge**[1,2], **Befikadu Bitewulign**[1],
**Abiyou Kiflie**[1], **Abera Biadgo**[1], **Haregeweyni Alemu**[1], **Yakob Seman**[3],
**Dorka Woldesenbet**[4], **Abiy Seifu Estifanos**[4], **Gareth Parry**[5], **Matthew Quaife**[6]

**1** Institute for Healthcare Improvement, Addis Ababa, Ethiopia, **2** Division of Global Health Equity, Brigham
and Women's Hospital, Boston, Massachusetts, United States of America, **3** Medical Service General
Directorate, Ministry of Health of Ethiopia, Addis Ababa, Ethiopia, **4** Department of Reproductive, Family and
Population Health, School of Public Health, Addis Ababa University, Addis Ababa, Ethiopia, **5** Department of
Plastic and Oral Surgery, Boston Children's Hospital, Boston, Massachusetts, United States of America,
**6** Department of Global Health and Development, London School of Hygiene and Tropical Medicine, London,
United Kingdom

* mehiret.ema@gmail.com

journal.pone.0272551

UNITED STATES

**Data Availability Statement:** "No - some
restrictions will apply" "The cross-sectional survey
data relevant to this study is restricted due to

## Abstract

### Background

Although Ethiopia has improved access to health care in recent years, quality of care
remains low. Health worker motivation is an important determinant of performance and
affects quality of care. Low health care workers motivation can be associated with poor
health care quality and client experience, non-attendance, and poor clinical outcome. Objec-
tive this study sought to determine the extent and variation of health professionals' motiva-
tion alongside factors associated with motivation.

### Methods

We conducted a facility based cross-sectional study among health extension workers
(HEWs) and health care professionals in four regions: Amhara, Oromia, South nations, and
nationalities people's region (SNNPR) and Tigray from April 15 to May 10, 2018. We sam-
pled 401 health system workers: skilled providers including nurses and midwives (n = 110),
HEWs (n = 210); and non-patient facing health system staff representing case team leaders,
facility and district heads, directors, and officers (n = 81). Participants completed a 30-item
Likert scale ranking tool which asked questions across 17 domains. We used exploratory
factor analysis to explore latent motivation constructs.

### Results

Of the 397 responses with complete data, 61% (95% CI 56%-66%) self-reported motivation
as "very good" or "excellent". Significant variation in motivation was seen across regions

concerns about participant identifiability. The data set contains indirect identifiers (job title, age, gender, time period working in role, time period working in health system) that might allow others, particularly those working in the hospitals & health centres covered, to determine the identify of a participant (or mistakenly attribute it to someone else). Data requests submitted through the LSHTM data repository (https://doi.org/10.17037/DATA.00001565) are sent to the project team and LSHTM Research Data Management Service for consideration. The LSHTM RDM Service acts as independent advisor for data access requests, directing them to the relevant ethics committee or other institutional body as appropriate. Data access requests can be submitted through the LSHTM data repository. Applicants may also email "researchdatamanagement@lshtm.ac.uk" with the DOI for the dataset being requested if preferred.".

**Funding:** This work was supported by IDEAS—Informed Decisions for Actions to improve maternal and newborn health (http://ideas.lshtm.ac.uk), which is funded through a grant from the Bill & Melinda Gates Foundation to the London School of Hygiene & Tropical Medicine. (Gates Global Health Grant Number: OPP1149259). The funder of this study had no role in the study's design or conduct, data collection, analysis or interpretation of results, writing of the paper, or decision to submit for publication.

**Competing interests:** The authors have declared that no competing interests exist.

with SNNPR scoring significantly lower on a five-point Likert scale by 0.35 points (P = 0.003). The exploratory factor analysis identified a three-factors: personal and altruistic goals; pride and personal satisfaction; and recognition and support. The personal and altruistic goals factor varied across regions with Oromia and SNNPR being significantly lower by 0.13 (P = 0.018) and 0.12 (P = 0.039) Likert points respectively. The pride and personal satisfaction factor were higher among those aged > = 30 years by 0.14 Likert scale points (P = 0.045) relative to those aged between 19-24years.

## Conclusions

Overall, motivation was high among participants but varied across region, cadre, and age. Workload, leave, and job satisfaction were associated with motivation.

## Introduction

Health worker motivation is an important determinant of health worker performance and, ultimately, health care quality [1, 2]. Although a range of factors affect health care quality, motivation is an important determinant of health worker effort, retention, and quality of provision, in addition to health care organization, resource availability, and other provider- and patient-related factors [2]. Motivation in the work context can be defined as an individual's degree of willingness to exert and maintain an effort towards organizational goals [3, 4]. Previous research has identified a number of factors that can affect health worker motivation, which can be broadly categorized in to three areas: social factors such as community expectations and social values; organizational factors such as resources and managerial support; and individual or process factors such as preferences or intrinsic attitudes [5, 6].

Several comparable studies have been conducted in sub-Saharan Africa. A study in Tanzania indicated that 45% of the individuals working in a primary health care unit were unsatisfied with their job. The reasons cited for dissatisfaction were low salaries, factors related to the working environment, and inadequate facilities for performing expected tasks [7]. A systematic review to assess motivation and retention of health care workers in developing countries indicated the importance of financial incentive on health workers motivation [8]. The studies indicated that health workers take pride and are motivated when they feel they could progress. Recognition by the employer and community was one of the most motivating factors for health workers alongside education and training opportunities. Low salaries were demotivating factors for health care provider [5, 8–14]. Low health care workers motivation is a major contributor to the poor health service quality and client experience, long waiting time, nonattendance, and unofficial fee charges and poor clinical outcome [15].

Three previous studies have assessed health worker motivation in Ethiopia quantitatively, though all were small in scale and focused on specific health worker cadres [3, 9, 16]. A study conducted in central Ethiopia revealed that the overall motivation of health professionals working in different hospitals of West Shewa Zone was reasonably high. Motivation was affected by financial incentives. The mean motivation score among health professionals who received monthly financial benefit was significantly higher than those who did not. Health professionals who had master's degrees and doctors had highest motivation. Professionals who worked for less than five years had less motivation [3]. A study done in public hospitals of West Amhara in Northwest Ethiopia revealed that the mean overall motivation scores were similarly high. Highly trained professionals, young age groups, and professionals who received

performance evaluations and professional development promotions were more motivated [16].

However, health care professionals' motivation working in public health centres in Gedeo Zone of South Nations and nationalities people's region (SNNPR) was very low. Lack of recognition and appreciation from their immediate supervisor or manager decreased their motivation. Work experience was a positive predictor of job motivation as work experience increased, job motivation increased [9]. A qualitative study indicated that Ethiopian primary health care workers commonly face work-related stress and experience features of burnout, which may contribute to the high turnover of staff and dissatisfaction of both patients and providers [17].

Though research has been done to identify factors associated with motivation in public hospitals and health centres in different regions of Ethiopia, there has been limited research conducted to explore the extent of motivation among Health extension workers (HEWs) working in the community and midlevel health care professionals across the four large regions of Ethiopia, or how motivation differs across cadres and regions. Thus, this study is sought to determine the extent and variation of health professionals' motivation alongside factors associated with motivation. We collected data from the four regions of Ethiopia, home to 81% of the country's population: Amhara; Oromia; SNNPR; and Tigray and accommodates most health facilities and health workers. These regions are highly diverse population and have different geographic characteristics improving representativeness [18].

## Materials and methods

### Study design

We conducted a facility based cross-sectional study among health extension workers (HEWs) and health care professionals in four regions: Amhara, Oromia, South nations, and nationalities people's region (SNNPR) and Tigray from April 15 to May 10, 2018.

### Tool development

We adapted a motivation tool developed and validated among community health workers (CHWs) in Uganda [19], making minor changes to wording to suit the Ethiopian context. The tool consisted of 17 questions. We added 8 additional questions from a health worker motivation evaluation conducted in Tanzania to explore extrinsic motivating factors in more depth [20]. Finally, we included 5 further questions relating to activities related to the quality improvement programme being implemented in our sample. The final tool is shown in Table 1.

### Sampling and data collection

We sampled 401 health system workers: skilled providers including nurses and midwives (n = 110), HEWs (n = 210); and non-patient facing health system staff representing case team leaders, facility and district heads, directors, and officers (n = 81). The survey was part of a baseline evaluation of a quality improvement (QI) program delivered by the Institute for Healthcare Improvement (IHI) in partnership with the Ministry of Health Ethiopia (MOH). Although the sampling frame of this study is based on the IHI program, data are from pre-intervention baseline data collection, and we do not expect motivation to have been influenced by the intervention at this point. The IHI program was implemented in 19 districts: 7 in Oromia, 5 in Amhara, 5 in SNNPR, and 2 in Tigray. Using a random number generator, we randomly selected one intervention district from each region (Jimma Town, Wogera, Chena, and

**Table 1. Motivation questions included in survey, source, and domain from April 15–May 10, 2018.**

| Domain | Wording | Source |
|---|---|---|
| Altruism | My work is important because I help people | [19] |
| Altruism | As long as I can do what I enjoy, I am not that concerned about exactly what income or awards I earn | QI-specific indicator |
| Community | I am respected in my community for the work I do | [19] |
| Income | I am strongly motivated by the income I can earn at work | [20] |
| Income | To be motivating, hard work must be rewarded with more status and money. | [20] |
| Income | My salary accurately reflects my skills and workload | [20] |
| Intention to leave | I intend to stop working in this role in the next 12 months | [19] |
| Job satisfaction | I am proud of the work I do | [20] |
| Job satisfaction | In general, I am satisfied with my role | [19] |
| Knowledge gain | I gain knowledge from being in this role | [19] |
| Knowledge gain | Training sessions that I attend are worthwhile and add benefit to my career path | QI-specific indicator |
| Motivation | At the moment I do not feel like working as hard as I can | [20] |
| Motivation | I feel like performing the duties required of me | [19] |
| Needs satisfaction | I am strongly motivated by the recognition I get from other people | [20] |
| Needs satisfaction | It is important that I do a good job so that the health system works well | [19] |
| Needs satisfaction | My job makes me feel good about myself. | [20] |
| Needs satisfaction | I feel it is not so important doing a good job if nobody else knows about it | QI-specific indicator |
| Organisational citizenship | I am willing to do more than is asked of me in my role | [19] |
| Organisational citizenship | Sometimes I do not understand why I am asked to do certain things, but I do them anyway | [19] |
| Organisational justice | The system of choosing who attends training sessions is fair | QI-specific indicator |
| Organisational justice | I do not have enough opportunities to attend training sessions to develop my career | QI-specific indicator |
| Outcome expectancy | I am keenly aware of the career goals I have set for myself | [20] |
| Outcome expectancy | If I do well at work, I will achieve my goals | [19] |
| Programme | I am proud to be working in my role | [19] |
| Programme commitment | I feel committed to my role | [19] |
| Resource availability | The health system provides everything I need to do my job properly | [19] |
| Self-efficacy | I can solve most problems I have at work if I work hard | [19] |
| Supervision | Suggestions made by people like me on how to improve their work are usually ignored by supervisors | [19] |
| Supervision | My supervisors and managers are supportive of me | [19] |
| Workload | I can complete all of the work I am expected to do | [19] |

All items had Likert scale response options where 1 = strongly agree, 2 = agree, 3 = neutral, 4 = disagree, 5 = strongly disagree.

Degua Tembien respectively). We added one additional randomly selected district in Amhara because Wogera would not have yielded 50 eligible respondents—our target for each region. We further purposively sampled two additional districts from Oromia and SNNPR (Bunno Bedelle and Chencha respectively) where qualitative evaluative work took place, to triangulate findings in a larger evaluation of IHI's QI program. Data collection was conducted by seven

research assistants who received one week training at the start of the data collection process and then were matched to their home regions where they have experience working in and speak local language to assist with community entry and mitigate language issues. The data quality was assured by using validated tools, trained data collectors, and conducting interviews in the local languages. The survey was piloted out of 19 district health office staff in December 2017. No changes were made to the survey between piloting and the final survey as it was understood well by participants, assessed through debriefing interviews after survey completion.

In each district, we mapped the hospital, all health centres and health posts, and approached the district health office for permission letters that was later obtained. In each hospital and health centre, we obtained a list of all eligible health care professionals and HEWs. We then randomly selected participants for interviews. In each district, we interviewed around 50 participants across a range of health worker and management cadres, including the heads or clinical directors of the district, each hospital, and each health centre. We interviewed around four maternal and child health care providers from the hospitals and two from each health centre, and around five HEWs from each health centre. A target sample size of 50 respondents per region was chosen, based on the primary research question of assessing changes in motivation as measured by Likert scale questions, in line with a rule of thumb in exploratory factor analysis that 50 participants per cluster is a reasonable sample size to detect differences across clusters [21].

In each hospital or health centre, we obtained a list of all eligible MNH providers and randomly selected participants for interviews. Their names were written in alphabetical order next to a column of randomly generated numbers and interviewers sequentially chose participants from the smallest random number upwards until the requisite number of participants was reached. If participants were not available, we sought to arrange interviews via phone and returned to the facility up to three times before classifying them as unreachable and selecting the next worker from the list. Data were entered on tablet computers using Open Data Kit software (www.opendatakit.org) and exported to STATA V.13.

## Data analysis

We categorized the responses according to sociodemographic factors using counts and percentages as appropriate.

To explore the underlying correlations and associations and identify factors within the survey items, we first re-coded the survey items from the 5-point Likert scale from poor to fair, good, very good and excellent to a continuous variable from 1 (poor) to 5 (excellent). Next, we used the re-coded items in an exploratory factor analysis. For the exploratory factor analysis, we first removed items from our list of 30 questions which had poor psychometric performance, removing items which had more than 10% missing data, items which were given the same score of over 80% of participants, and items which did not load on any factors up to a level of 0.4 in initial factor analysis. We used a threshold of 0.4 to assume a strong relationship with a factor, and the optimal number of factors was established through a scree test and multiple runs [22, 23]. We used maximum likelihood ProMax oblique rotation to reduce the number of variables with high loadings and to allow factors to be correlated. Construct validity was indicated by loading at least three items per factor and absence of substantive cross-loading.

We explored the association of overall motivation and with the motivation factors identified with overall job satisfaction and demographic and structural factors including gender, location, cadre, age, perceived gross salary, work experience, using univariate and multivariate ordinary least squares regression models, and show ordered logit model results in the S1 & S2

Appendices. Variables having p value $\leq 0.2$ in the bivariate analysis were fitted into a multivariable regression model to control the effects of confounding. Normality assumptions were checked by Schapiro—Francia W tests, and variance inflation factor estimates were generated for regressors [24]. Average job satisfaction was assessed by re-coding the 5-point Likert scale ranging from least satisfied with their job (1) to most satisfied (5) as a continuous variable.

## Ethical considerations

Written informed consent was obtained from all participants. The study was undertaken with ethical approval from the Observational Research Ethics Committee of the London School of Hygiene and Tropical Medicine (Ref: 14429) and a program evaluation waiver from the Ethics Committee of the Ethiopian Public Health Association (Ref: EPHA/OG/2380).

# Results

## Sociodemographic characteristics

Of 401 people surveyed, 397 responded complete giving a response rate of 99%. Most of the respondents were between 25–30 years of age (232, 58.44%) and female (288, 72.73%). Two hundred five (51%) of respondents had greater than 4 years of work experience, and (208, 52.39%) of the respondents were HEWs shown in Table 2.

Fig 1 plots the responses to each motivational task, where 1 represents "strongly agree" and 5 represents "strongly disagree", and blue dots represent items where a lower score is a priori

**Table 2. Background characteristics of participants from four regions, (n = 397), April 15–May 10, 2018.**

| Variables | Number (%) |
|---|---|
| Region | |
| Amhara | 81 (20.40) |
| Oromia | 105 (26.45) |
| SNNPR | 137 (34.51) |
| Tigray | 74 (18.64) |
| Age (Years) | |
| 19–24 | 90 (22.67) |
| 25–30 | 232 (58.44) |
| >30 | 75 (18.89) |
| Job title | |
| Health extension workers | 208 (52.39) |
| Care provider (Nurse, Midwife, health officer…) * | 109 (27.46) |
| Case team Leaders | 63 (15.71) |
| Other (facility and district heads, directors, and officers) | 17 (4.28) |
| Gender | |
| Male | 109 (27.46) |
| Female | 288 (72.54) |
| Work experience | |
| <6month | 56 (14.11) |
| 6month-1year | 40 (10.08) |
| 1-2years | 42 (10.58) |
| 2-4years | 54 (13.60) |
| >4years | 205 (51.64) |

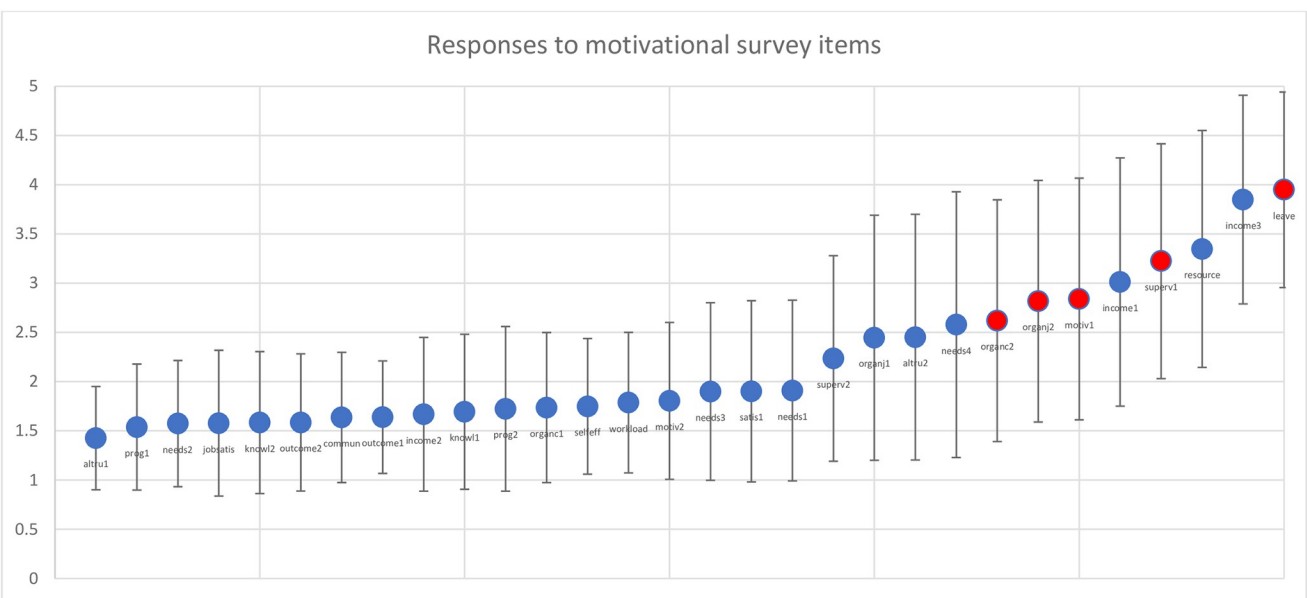

**Fig 1. Plot of responses to motivational survey items among participants from four regions of Ethiopia, April 15–May 10, 2018.**

better (e.g., "I am respected in my community for the work I do") and red dots where a lower score is worse (e.g., "I intend to stop working in this role in the next 12 months").

## Motivation construction—Exploratory factor analysis

We ran a factor analysis and ultimately found that the three-factor model fit the data best. We removed 11 items which did not load to 0.4 on any factor. We calculated the final factor score by multiplying the items by the factor score and summing. Table 3 shows the factor loadings. In addition to the factor loading above 0.4, we used eigen values >1 as a criterion to reduce the number of factors into three. We labelled the three factors as personal and altruistic goals; pride and personal satisfaction; and recognition and support. The minimum and maximum score for personal and altruistic goals was 1 and 4.85; for pride and personal satisfaction it was 1 and 4.17; and for recognition and support, it was 1 and 4.5.

## Motivation level

Of the total respondents, 61% were motivated, responding "very good" or "excellent" (95% CI 57%–66%). The most motivating factor mentioned by 70% of the participants was the opportunity to improve health. The most demotivating factor mentioned by 29% participants was workload (95% CI 34%–39%).

**Factors related to overall motivation.** Table 4 shows the results of the regression model with overall motivation as the outcome variable and several explanatory factors fitted.

The overall motivation score mean was 2.18 out of 5 (95% CI 2.10, 2.27; P = 0.001). Average job satisfaction was strongly associated with higher motivation (P = 0.001). We found variation in motivation across regions, where participants from SNNPR reported lower motivation than the Amhara region by 0.35 (95% CI 0.12, 0.59; P = 0.003). Participants who have a medium workload (meaning that they have enough time to complete duties) were also less motivated than participants who have a light workload by 0.48 points (95% CI -0.90, -0.06; P = 0.024), though high workload was not significantly different. There was no significant variation in overall motivation with job title and perceived fairness of gross salary.

**Table 3. Exploratory factor analysis results showing the factor loadings by individual items April 15–May 10, 2018.**

| Variable | Factor 1: Personal and altruistic goals | Factor 2: Pride and personal satisfaction | Factor 3: Recognition and support | Uniqueness |
|---|---|---|---|---|
| My work is important because I help people | 0.65 | | | 0.44 |
| I am respected in my community for the work I do | 0.58 | | | 0.51 |
| I am keenly aware of the career goals I have set for myself | 0.57 | | | 0.50 |
| I feel committed to my role | 0.56 | | | 0.44 |
| If I do well at work, I will achieve my goals | 0.54 | | | 0.57 |
| I am willing to do more than is asked of me in my role | 0.47 | | | 0.68 |
| I can solve most problems I have at work if I work hard | 0.37 | | | 0.70 |
| I can complete all the work I am expected to do | | 0.41 | | 0.68 |
| I feel like performing the duties required of me | | 0.53 | | 0.55 |
| I am proud of the work I do | | 0.57 | | 0.50 |
| I am proud to be working in my role | | 0.65 | | 0.43 |
| In general, I am satisfied with my role | | 0.57 | | 0.64 |
| My job makes me feel good about myself. | | 0.54 | | 0.57 |
| It is important that I do a good job so that the health system works well | | | 0.48 | 0.51 |
| Training sessions that I attend are worthwhile and add benefit to my career path | | | 0.49 | 0.59 |
| To be motivating, hard work must be rewarded with more status and money. | | | 0.58 | 0.57 |
| I am strongly motivated by the recognition I get from other people | | | 0.44 | 0.62 |
| I gain knowledge from being in this role | | | 0.62 | 0.38 |
| My supervisors and managers are supportive of me | | | 0.36 | 0.73 |

Factor 1 interpreted to represent personal and altruistic goals. Factor 2 interpreted to represent pride and personal satisfaction. Factor 3 interpreted to represent recognition and support.

## Factor 1: Personal and altruistic goals

The first factor identified items which related to personal success and goal setting and altruism. We refer to this factor as "personal and altruistic goals". The mean response for personal and altruistic goals was 1.61 Likert points (95% CI 1.56, 1.66; P = 0.001). From the results of the regression analysis summarized in Table 5, average job satisfaction was strongly associated with higher personal and altruistic goals score (P = 0.001). Variation in personal and altruistic goals was seen among regions, where participants from Oromia and SNNPR had lower personal and altruistic goal scores than Amhara region by 0.13 Likert points (95% CI -0.25, -0.02; P = 0.018) and 0.12 Likert points (95% CI -0.23, -0.01; P = 0.039) respectively. Average leave (days out of work) was another significant factor associated with personal and altruistic goals, where an increase of 1 was associated with a decrease in personal and altruistic goals of 0.08 Likert points (95% CI -0.12, -0.04; P = 0.001). Age, gender, and perceived fairness of salary had no significant association with personal and altruistic goals.

## Factor 2: Pride and personal satisfaction

The second factor consisted of variables relating to the pride and personal satisfaction respondents experienced in their jobs, and the satisfaction their jobs gave them. The mean score of pride and personal satisfaction was 1.78 Likert points (95% CI 1.72, 1.84; P = 0.001). A regression analysis in Table 5 indicated that pride and personal satisfaction score vary among age

**Table 4. Factors associated with overall motivation among participants form four regions April 15–May 10, 2018.**

| Factor | Coef. | 95% CI | P-value |
|---|---|---|---|
| Region | | | |
| Amhara | Reference | | |
| Oromia | 0.19 | (-0.04, 0.42) | 0.110 |
| SNNPR | -0.35 | (-0.59, -0.12) | 0.003 |
| Tigray | -0.05 | (-0.32, 0.21) | 0.685 |
| Job title | | | |
| HEW | Reference | | |
| Health care providers | 0.13 | (0.32, -0.05) | 0.161 |
| Case team Leaders | -0.19 | (0.03, -0.41) | 0.096 |
| Other (facility and district heads, directors, and officers) | -0.23 | (0.17, -0.64) | 0.269 |
| Workload | | | |
| Light: more than enough time to complete duties | Reference | | |
| Medium: enough time to complete duties | -0.48 | (-0.90, -0.06) | 0.024 |
| Heavy: barely enough time to complete duties | -0.29 | (0.11, 0.70) | 0.163 |
| Perceived gross salary fair | | | |
| Very fair | Reference | | |
| Quite fair | 0.00 | (-0.79, 0.80) | 0.995 |
| Neither fair nor unfair | 0.29 | (-0.54, 1.12) | 0.488 |
| Quite unfair | 0.10 | (-0.69, 0.90) | 0.798 |
| Very unfair | -0.04 | (-0.87, 0.78) | 0.914 |
| Average job satisfaction | 0.23 | (0.13, 0.34) | 0.001 |

groups. There were significantly high scores among those older than 30 years old by 0.14 Likert points (95% CI 0.01, 0.28; P = 0.045) as compared to those between the ages of 19–24 years old. Likewise, as average leave (time out of work) increased by 1, pride and personal satisfaction score decreased by 0.06 Likert points (95% CI -0.09, -0.02; P = 0.004). Another factor affecting pride and personal satisfaction was average job satisfaction. Average job satisfaction is strongly associated with high pride and personal satisfaction (P = 0.001). Gender, Region, and work experience did not have significant association with pride and personal satisfaction.

## Factor 3: Recognition and support

Factor 3 synthesised survey items focusing on the recognition and support those participants received from colleagues and seniors. The mean score of recognition and support was 1.77 (95% CI 1.72, 1.83; P = 0.001). From the regression analysis indicated in Table 5, factors associated with recognition and support were job title, region, leave (time out of work) and job satisfaction. Recognition and support scores were high among health care providers as compared to HEWs by 0.12 Likert points (95% CI 0.001, 0.25; P = 0.053). Compared to Amhara region, recognition and support scores were significantly low among SNNPR by 0.15 Likert points (95% CI -0.32, -0.03; P = 0.017) and Oromia region by 0.17 Likert points (95% CI -0.29, -0.01; P = 0.034). Another factor affecting recognition and support was job satisfaction. Average job satisfaction is strongly associated with higher recognition and support score (P = 0.001). However, as average leave days increased by one, recognition and support scores significantly decreased by 0.09 Likert points (95% CI -0.14, -0.04; P = 0.001). Age, work experience, gender, and perception of fairness of salary has no significant association with recognition and support.

**Table 5. Association between motivation factors and demographic and structural factors among participants from four regions April 15–May 10, 2018.**

| Factor | Factor 1: personal and altruistic goals | | | Factor 2: pride and personal satisfaction | | | Factor 3: recognition and support | | |
|---|---|---|---|---|---|---|---|---|---|
| | Coef. | 95% CI | P-value | Coef. | 95% CI | P-value | Coef. | 95% CI | P-value |
| Gender | | | | | | | | | |
| Male | Reference | | | Reference | | | Reference | | |
| Female | 0.01 | (-0.08, 0.09) | 0.896 | -0.02 | (-0.12, 0.08) | 0.720 | -0.01 | (-0.17, 0.14) | 0.863 |
| Region | | | | | | | | | |
| Amhara | Reference | | | Reference | | | Reference | | |
| Oromia | 0.13 | (-0.02, -0.25) | 0.018 | 0.00 | (-0.11, 0.11) | 0.969 | -0.15 | (-0.29, -0.01) | 0.034 |
| SNNPR | 0.12 | (-0.01, -0.23) | 0.039 | - 0.10 | (-0.21, 0.01) | 0.064 | -0.17 | (-0.32, -0.03) | 0.017 |
| Tigray | 0.04 | (-0.17, 0.08) | 0.523 | 0.08 | (-0.04, 0.2) | 0.175 | 0.08 | (-0.07, 0.24) | 0.290 |
| Age | | | | | | | | | |
| 19–24 | Reference | | | Reference | | | Reference | | |
| 25–30 | 0.04 | (-0.14,0.5) | 0.403 | 0.07 | (-0.02, 0.19) | 0.180 | -0.02 | (-0.16, 0.10) | 0.710 |
| >30 | 0.01 | (-0.01, 0.13) | 0.915 | 0.14 | (0.01, 0.28) | 0.045 | 0.04 | (-0.13, 0.21) | 0.633 |
| Perceived gross salary fair | | | | | | | | | |
| Very fair | Reference | | | | | | Reference | | |
| Quite fair | -0.27 | (0.10, -0.65) | 0.155 | | | | -0.15 | (-0.61, 0.30) | 0.514 |
| Neither fair nor unfair | -0.27 | (0.12, -0.66) | 0.181 | | | | 0.01 | (-0.47, 0.48) | 0.976 |
| Quite unfair | -0.21 | (0.17, -0.59) | 0.277 | | | | 0.01 | (-0.45, 0.46) | 0.973 |
| Very unfair | -0.26 | (0.13, -0.65) | 0.199 | | | | 0.09 | (-0.39, 056) | 0.712 |
| Work experience | | | | | | | | | |
| <6month | | | | Reference | | | | | |
| 6month-1year | | | | -0.02 | (-0.18, 0.14) | 0.824 | -0.35 | (-0.83, 0.12) | 0.147 |
| 1-2years | | | | -0.00 | (-0.16, 0.15) | 0.971 | -0.29 | (-0.76, 0.18) | 0.227 |
| 2-4years | | | | 0.11 | (-0.37, 0.26) | 0.145 | -0.34 | (-0.80, 0.12) | 0.151 |
| >4years | | | | -0.06 | (-0.18, 0.07) | 0.366 | -0.39 | (-0.85, 0.07) | 0.099 |
| Job title | | | | | | | | | |
| HEW | | | | | | | Reference | | |
| Healthcare provides | | | | | | | 0.12 | (0.00, 0.25) | 0.053 |
| Case team Leaders | | | | | | | 0.02 | (-0.17, 0.20) | 0.862 |
| Other (facility and district heads, directors, and officers) | | | | | | | -0.01 | (-0.28, 0.25) | 0.929 |
| Average job satisfaction | 0.34 | (0.29, 0.39) | 0.001 | 0.06 | (0.09, 0.02) | 0.004 | 0.09 | (0.14, 0.04) | 0.001 |
| Average leave | -0.08 | (-0.04, -0.12) | 0.001 | 0.56 | (0.50, 0.61) | 0.001 | 0.35 | (0.29, 0.41) | 0.001 |

## Discussion

Our study results indicated that more than 60% of HEW and health workers were motivated to do their job. Key factors identified to influence motivation were region, age, job title, work experience, job satisfaction, and leave days. The motivation domains identified in a three-

factor analysis was effort in personal and altruistic goals; pride and personal satisfaction; and recognition (financial and managerial) support.

Our finding of high overall motivation is consistent with several other studies. For example, a study conducted in West Amhara, Northwest Ethiopia showed that 59% of care providers working in hospitals were motivated [16]. In West Shewa Zone of Oromia region, the overall mean motivation score of health professionals from three hospitals was 64% [3]. In this study, motivation varied significantly from region to region, which previous smaller studies have not been able to explore. This variation is consistent with a study from Kenya that also found motivational factors varied from region to region [11]. SNNPR had the lowest motivation, consistent with a study done among health professionals in SNNPR's Gedeo zone where only 17% of respondents were highly motivated. This could need further exploratory studies [9].

Personal and altruistic goals of health providers were significantly lower among participants from Oromia and SNNPR compared to those from the Amhara region. Average job satisfaction is strongly associated with high efforts in personal and altruistic goal.

Motivation related to pride and personal satisfaction was significantly high among participants older than 30 years of age as compared to those 24 years and younger. This study was consistent with studies from Central Ethiopia and South Ethiopia where age was a major predictor of motivation; as age increased, motivation to do their job increased [9, 12]. Similar studies in Burkina Faso, Ghana and Tanzania showed that age has a significant effect on motivation [25]. Pride and personal satisfaction factor were significantly high among participants whose job satisfaction was high as of different studies in Ethiopia explains [26].

Recognition and support were significantly low among participants from SNNPR and Oromia region and was associated with low motivation. This could be due to instability within the system or the lack of capacity to supervise the health care workers located in a large geographical area. Among participants with increased average leave, recognition support was significantly low. This may explain why participants who were taking leave more were less motivated. Recognition and support were significantly higher among health care providers than among HEWs. Having more recognition (financial and managerial) support was a positive predictor for job satisfaction and motivation [8, 27, 28].

A strength of this study was that data was collected from four (Amhara, Oromia, SNNPR, Tigray) major regions of Ethiopia where most of the country's population live and the regional variation of motivation was studied at the same time, and the health care workers and health extension workers motivation were compared, in addition this study addressed computing multiple associated factors with the overall motivation and three motivation factors identified during factor analysis. Nevertheless, limitations remain. Motivation was self-reported and may therefore be subject to acceptability biases in the face-to-face interviews conducted. Knowledge that interviews were conducted by interviewers from a public university and evaluating an IHI programme may have biased responses. The skilled care provider sample was too small to conduct a subgroup analysis among specific groups of interest such as doctors or midwives. Region-level analyses may have been underpowered to variation in motivation, particularly in the presence of heterogeneity in motivation within regions, for example by cadre, or other unobserved variables. The sampling strategy was not optimized to account for or detect potentially important heterogeneity between regions, urban and rural areas, or QI programme areas, which may further reduce power to detect across-region variation. Qualitative work could have been conducted alongside quantitative data collection to understand why variation exists in motivation between regions and personal characteristics. Finally, we were unable to assess the link between health worker motivation and the quality of care provided which remains under-researched.

## Conclusion

Motivation was high among the respondents from the four regions. Key factors associated with motivation were region, age, job title, workload, job satisfaction, long leave days and work experience. Further studies are needed to explore the reason for variation in motivation across regions and job title.

## Supporting information

**S1 Questionnaire. Questionnaire used for data collection in ODK file was prepared in both local (Amharic) and English language.**
(XLSX)

**S1 Appendix. Table A1.** Results from ordered logit model showing factors associated with overall motivation among participants form four regions April 15- May10, 2018.
(PDF)

**S2 Appendix. Table A2.** Results from ordered logit model showing association between motivation factors and demographic and structural factors among participants from four regions April 15- May10,2018.
(PDF)

## Acknowledgments

We are grateful to all participants for their time in completing various study components. Excellent research assistance in data collection was provided by Teklit Grum, Tsegaab Temesgen, Zelalem Tedla, Saba Shiferaw, Duresa Endale, Abeba Hailemelekot, Worknesh Daba, Hasna Musema, Selamawit Herpa, Tigist Abera, Samuel Ayele, and Aniley Dagife. We thank our colleagues from Addis Continental Institute of Public Health who provided insight and expertise that greatly assisted with the analysis and write-up of this research. We are especially indebted to Prof. Alemayehu Worku and Dr Walelign Worku for their tireless support in the analysis of the data. We are also grateful to Jane Roessner for her valuable comments on an earlier version of the manuscript and Naomi Fedna for proofreading the final version to fit journal requirement.

## Author Contributions

**Conceptualization:** Mehiret Abate, Zewdie Mulissa, Gareth Parry, Matthew Quaife.

**Data curation:** Mehiret Abate, Gareth Parry, Matthew Quaife.

**Formal analysis:** Mehiret Abate, Matthew Quaife.

**Investigation:** Zewdie Mulissa, Befikadu Bitewulign, Abera Biadgo, Haregeweyni Alemu, Dorka Woldesenbet, Abiy Seifu Estifanos.

**Methodology:** Mehiret Abate, Gareth Parry, Matthew Quaife.

**Project administration:** Zewdie Mulissa, Hema Magge, Befikadu Bitewulign, Abiyou Kiflie, Abera Biadgo, Haregeweyni Alemu.

**Resources:** Abiyou Kiflie, Matthew Quaife.

**Supervision:** Hema Magge, Dorka Woldesenbet, Gareth Parry, Matthew Quaife.

**Writing – original draft:** Mehiret Abate.

**Writing – review & editing:** Zewdie Mulissa, Befikadu Bitewulign, Abiyou Kiflie, Abera Biadgo, Haregeweyni Alemu, Yakob Seman, Dorka Woldesenbet, Abiy Seifu Estifanos, Gareth Parry, Matthew Quaife.

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
