## [Decision Letter · Decision Letter 0]

30 Jun 2021

PONE-D-21-09235

Key factors influencing motivation among health extension workers and health care professionals in four regions of Ethiopia: a cross-sectional study.

PLOS ONE

Dear Dr. Adillo,

Thank you for submitting your manuscript to PLOS ONE. After careful consideration, we feel that it has merit but does not fully meet PLOS ONE’s publication criteria as it currently stands. Therefore, we invite you to submit a revised version of the manuscript that addresses the points raised during the review process.

The submission has now been reviewed and I have received the referees’ evaluation of your paper. As you can see, the reviewers find the theme of your manuscript interesting but point to shortcomings and weaknesses that need to be addressed and remedied. When resubmitting please indicate how the revised version addresses all the referees’ concerns and concomitant suggestions.

I also have some comments that I hope can improve the current version of the paper:

1) raw 127: you mention the "woredas", I am suspecting it is something like regions? Please either give the interpretation for international readers that are not familiar with local terms or substitute with the international term.

2) raws 175-176: you mention that you utilise regression models. Please elaborate more. Do you estimate OLS models (I am suspecting yes, based on my comment below) and why you choose the OLS regression since your dependent is qualitative?

3) Raws 179-181: you state that the job satisfaction scores are recoded to transform the variable to a continuous one (if I understand correctly). How do you do that and what is the rationale to justify this transformation?

4) Raws 240-242: you state that "... where an increase of 1 in job 241 satisfaction was associated with an increase in motivation of 0.23 points". I would discourage you from using such strong assumptions. For one, the models estimated do not necessarity imply causation and secondly, both variables are qualitative so such an interpration has no logical meaning. I would advise estimating logit/ordered logit models and assess the probability to respond a higher/lower score base on certain characteristics of respondents, since it would make more sense to do that. Please remove such quantitative interpretations from the results and the discussion.

5) Please interpret your findings with caution: state the problems of establishing causality and remove statements that suggest otherwise (for example rows 316-317).

We look forward to receiving your revised manuscript.

Kind regards,

Athina Economou

Academic Editor

PLOS ONE

Journal Requirements:

2. Please include additional information regarding the survey or questionnaire used in the study and ensure that you have provided sufficient details that others could replicate the analyses. For instance, if you developed a questionnaire as part of this study and it is not under a copyright more restrictive than CC-BY, please include a copy, in both the original language and English, as Supporting Information. Moreover, please include more details on how the questionnaire was pre-tested, and whether it was validated.

4. Please upload a copy of Supporting Information which you refer to in your text on page 24.

Reviewers' comments:

Reviewer's Responses to Questions

**Comments to the Author**

1. Is the manuscript technically sound, and do the data support the conclusions?

Reviewer #1: Partly

Reviewer #2: Yes

2. Has the statistical analysis been performed appropriately and rigorously? 

Reviewer #1: Yes

Reviewer #2: Yes

3. Have the authors made all data underlying the findings in their manuscript fully available?

Reviewer #1: Yes

Reviewer #2: No

4. Is the manuscript presented in an intelligible fashion and written in standard English?

Reviewer #1: Yes

Reviewer #2: Yes

5. Review Comments to the Author

Reviewer #1: It is a commendable study to have collected data from four different regions of Ethiopia and different cadres of health workers, including HEWs. The addition of the following information would improve the quality of this manuscript.

1. references

In the Introduction/Methods sections, it is better to cite the reference (e.g. line 60 of page 3, line 66 of page 4 [to add additional reference as you state "three studies" in the beginning], line 100 of page 5, and Source column of Table 1 of page 6 [rather than describing which study, cite the reference number])

2. name of the location: "Degua Temben" line 129 on page 8, might be Degua "Tembien"

3. Sampling

One of the biggest concerns of this manuscript was the sampling. Please kindly respond to the following points:

a. Does the sample size for this study have enough statistical power to discern the regional difference of motivation among four regions, especially having Tigray only 50 participants.

b. Please state how the target of 50 respondents was set.

c. Did the authors consider cluster random sampling taking population size, rural/urban differences, intervention/non-intervention areas of IHI into consideration? If not, why not?

Reviewer #2: This is the manuscript titled "Key factors influencing motivation among health extension workers and health care professionals in four regions of Ethiopia: a cross-sectional study." by Mehiret A. et al.

Firstly, I would like to thank the authors for this work. This is an important study that attempts to address Key factors influencing motivation among health extension workers and health care professionals in four regions of Ethiopia. The study sheds light on the challenges of ensuring quality care through motivated health extension workers and health care professionals. All countries worldwide have signed up to the United Nations Sustainable Development Goals (UNSDGs) and have committed to achieving universal health coverage, including financial risk protection, access to quality essential healthcare services, and access to safe, adequate, quality, affordable essential medicines and vaccines for all. However, health care coverage in Ethiopia is still a big concern, which requires addressing all components. Therefore, I am glad to be a reviewer of this work.

As a general comment, the authors should improve the language, introduction, and discussion section.

Specific comments

Abstract

- Background: The authors need to justify more the reason that initiates them to study this topic. Make sure to show the gap clearly, and include the objective also.

- Method: the authors should explain the critical components of an abstract such as study period, study setting,

Introduction:

- The introduction was written pretty well but still needs to show the consequences of inadequate health professionals and health extension worker motivation on the population health in general, such as death, a dropout from service, prolonged hospitalization, poor attitude to health facilities visit and soon.

Methods

- Line page 10: check for typo-error such as … established through a “Scree” test and multiple…

- Page 10, on lines 179 to 181, the authors mentioned that “the average job satisfaction was assessed by re-coding the 5-point Likert scale ranging from most satisfied with their job (1) to least satisfied (5) as a continuous variable.” Why did they need to consider scale one as most satisfied and scale five least? The revers will be more logical. Can you please explain this? Most studies and scholars use 5 as the most satisfied, while 1 is the least satisfied.

Result sections

1. Table titles are not self-explanatory. Table/figure titles must be self-explanatory, which means every audience should be understood what the figure/table contains.

2. The number of the total participants is not consistently explained in the document. For example, the authors told us that only 397 participants were provided their responses. However, they explained 401 respondents in the table. Similarly, the authors should re-check that the sum of all participants is similar to the total respondents. E.g., in Table 2: region, the sum of 107, 106, 137, 50 is 400, which is not equal to the total participants explained by the title. Same table, data for age: the sum of 90, 232, 75 is …. Check other also.

3. Page 16, line 234/5… most demotivating factor mentioned by 29% … add the confidence interval.

4. What are your criteria for selecting a reference group in the regression analysis?

Discussion section

1. What is the reason or reference to say motivation among HEW is high?

2. What is the possible reason health extension workers get motivated in the Amhara region compared to the SNNP? This might be good to scale up to other regions.

Recommendation

- Based on your finding, what do you suggest/recommend for the concerned body?

6. PLOS authors have the option to publish the peer review history of their article (what does this mean?). If published, this will include your full peer review and any attached files.

Reviewer #1: No

Reviewer #2: **Yes: **Cheru Tesema Leshargie

---

## [Author Response · Author response to Decision Letter 0]

11 Oct 2021

September 3, 2021

Manuscript PONE-D-21-09235 

Response to reviewers 

Dear Editor in Chief

Thank you so much for the opportunity you have given us to re-submit a revised manuscript entitled “Key factors influencing motivation among health extension workers and health care professionals in four regions of Ethiopia: a cross sectional study.” for consideration for publication in PLOS ONE.

We appreciate the time and effort you and the reviewers dedicated to providing feedback to our manuscript and are grateful for the insightful comments on and valuable improvements made to our paper. 

We have incorporated most of the suggestions made by the reviewers. Those changes are highlighted with yellow color within the revised manuscript with track changes. Please see below a blue colored point-by-point response to the reviewers’ comments and concerns. Please note that all page numbers and line numbers refer to the revised manuscript.

Thank you so much once again! 

Reviewers’ comments to the Authors 

Comment from academic editor

1. Row 127: you mention the "woredas", I am suspecting it is something like regions? Please either give the interpretation for international readers that are not familiar with local terms or substitute with the international term.

Authors response: thank you. The author substituted “Woreda” with known equivalent term “District” as indicated in line 129,130,132,145,146,148,150.

Comment from academic editor

2. Row 175-176: you mention that you utilise regression models. Please elaborate more. Do you estimate OLS models (I am suspecting yes, based on my comment below) and why you choose the OLS regression since your dependent is qualitative?

Authors response: thank you. Yes, the Authors used OLS models to explore determinants of motivation level. We have now changed methods text to read (additions in bold): “We explored the association of overall motivation and with the motivation factors identified with overall job satisfaction and several demographic and structural factors including gender, location, cadre, age, perceived gross salary, work experience, using univariate and multivariate ordinary least squares regression models, and show ordered logistic regression model results in the appendix 1&2 mentioned in line 186 and supporting file S2&3 as in line 524&516.

Ordinary least squares models are able to give unbiased coefficients for ordered categorical outcomes, though we accept the point that interpretation is difficult, and this is non-standard in some fields. As above, therefore, we have fitted ordered logistic regression models to the data and present results in the appendix. The results of these models are consistent with OLS results with respect to sign, significance, and magnitude of independent variable coefficients. 

Comment academic editor

3. Rows 179-181: you state that the job satisfaction scores are recoded to transform the variable to a continuous one (if I understand correctly). How do you do that and what is the rationale to justify this transformation?

Author response: We did not transform the variables; we simply took categorical Likert Scale responses ranging from most satisfied in their job (5) to least satisfied (1) and used this as dependent variable (as described in line 190-191).

Comment from academic editor

4. Rows 240-242: you state that "... where an increase of 1 in job 241 satisfaction was associated with an increase in motivation of 0.23 points". I would discourage you from using such strong assumptions. For one, the models estimated do not necessarily imply causation and secondly, both variables are qualitative, so such an interpretation has no logical meaning. I would advise estimating logit/ordered logit models and assess the probability to respond a higher/lower score base on certain characteristics of respondents, since it would make more sense to do that. Please remove such quantitative interpretations from the results and the discussion.

Author response: Thank you for pointing this out, and we agree interpretation is difficult here. We have changed these results to describe sign and significance rather than magnitude as suggested on the following lines: 

line 251-253 reads as “Average job satisfaction was strongly associated with higher motivation (P=0.001).”

line 273-285 reads as “From the results of the regression analysis summarized in Table 5, average job satisfaction was strongly associated with higher personal and altruistic goals score (P=0.001).”

line 294-296 reads as “Average job satisfaction is strongly associated with high pride and personal satisfaction (P=0.001)”

line 309-311 reads as “Average job satisfaction is strongly associated with higher recognition and support score (P=0.001).”

Comment from academic editor

5. Please interpret your findings with caution: state the problems of establishing causality and remove statements that suggest otherwise (for example rows 316-317). 

Author response: Thank you for pointing this out, and as above we have changed this language throughout. Sentences corrected based on the reviewer comments as in line 331-333: reads as “Average job satisfaction is strongly associated with high efforts in personal and altruistic goal.”

Comment from reviewer #1

It is a commendable study to have collected data from four different regions of Ethiopia and different cadres of health workers, including HEWs. The addition of the following information would improve the quality of this manuscript.

Author response: Thank you for reviewing the manuscript, and for your comments which have improved the paper.

Comment from Reviewer #1

1. References

In the Introduction/Methods sections, it is better to cite the reference (e.g. line 60 of page 3, line 66 of page 4 [to add additional reference as you state "three studies" in the beginning], line 100 of page 5, and Source column of Table 1 of page 6 [rather than describing which study, cite the reference number])

Authors response: thank you for pointing this out. Reference included in the introduction and Method section, and on table 1 source column as on line 67, 76,109, 115&16 table 1 of page 7 line.

Comment from Reviewer #1

2. Name of the location: "Degua Temben" line 129 on page 8, might be Degua "Tembien"

Author response: thank you it is a typing error corrected as Degua “Tembien” as on line 131 

Comment from Reviewer #1

3. Sampling One of the biggest concerns of this manuscript was the sampling. Please kindly respond to the following points:

a. Does the sample size for this study have enough statistical power to discern the regional difference of motivation among four regions, especially having Tigray only 50 participants?

Author response: Thank you for pointing this out, this is a good point. We found this as a typing error where the participants from Tigray were74 as shown in page 13 of table 2, in addition we have added the following text in the limitations section of the discussion in line 356-359:

“Region-level analyses may have been underpowered to variation in motivation, particularly in the presence of heterogeneity in motivation within regions, for example by cadre, or other unobserved variables.”

b. Please state how the target of 50 respondents was set.

Author response: Thank you for allowing us to clarify this. We have added the following to the methods section as shown in line153-156:

“A target sample size of 50 respondents per region was chosen, based on the primary research question of assessing changes in motivation as measured by Likert scale questions, in line with a rule of thumb in exploratory factor analysis that 50 participants per cluster is a reasonable sample size to detect differences across clusters (20).”

c. Did the authors consider cluster random sampling taking population size, rural/urban differences, intervention/non-intervention areas of IHI into consideration? If not, why not?

Author response: Thank you for pointing this out. Cluster random sampling was not possible due to operational constraints, but the reviewer is correct to suggest that this would have allowed us to consider variation by potentially important observable characteristics in the sampling design. We have added the following to the limitations section to acknowledge this as shown in line 359-361:

“The sampling strategy was not optimized to account for or detect potentially important heterogeneity between regions, urban and rural areas, or QI program areas, which may further reduce power to detect across-region variation.”

Comment from reviewer#2

This is the manuscript titled "Key factors influencing motivation among health extension workers and health care professionals in four regions of Ethiopia: a cross-sectional study." by Mehiret A. et al.

Firstly, I would like to thank the authors for this work. This is an important study that attempts to address Key factors influencing motivation among health extension workers and health care professionals in four regions of Ethiopia. The study sheds light on the challenges of ensuring quality care through motivated health extension workers and health care professionals. All countries worldwide have signed up to the United Nations Sustainable Development Goals (UNSDGs) and have committed to achieving universal health coverage, including financial risk protection, access to quality essential healthcare services, and access to safe, adequate, quality, affordable essential medicines and vaccines for all. However, health care coverage in Ethiopia is still a big concern, which requires addressing all components. Therefore, I am glad to be a reviewer of this work.

As a general comment, the authors should improve the language, introduction, and discussion section. 

Author response: Thank you. we are very much grateful to the reviewers for their close reading of the paper and constructive comments which we believe to improve our paper. 

Comment from reviewer #2

1. Background: The authors need to justify more the reason that initiates them to study this topic. Make sure to show the gap clearly and include the objective also. 

Author Response: thank you for pointing this out. The reviewer is correct, and we have included the gap and objective. 

The revised text on background section of Abstract on line 24-27 of Page2 reads as “Low health care workers motivation can be associated with poor health care quality and displeasure to clients, long patient waiting time, non-attendance, and poor clinical outcome. Thus, this study sought to determine the extent and variation of health professionals’ motivation alongside factors associated with motivation.”

Comment from reviewer #2

2. Method: the authors should explain the critical components of an abstract such as study period, study setting, 

Author response: Thank you for pointing this out. Study time and setting incorporated on the method section of the Abstract line 28-30 of page2

“We conducted a facility based cross-sectional study among health extension workers (HEWs) and health care professionals in four regions: Amhara, Oromia, South nations, and nationalities people’s region (SNNPR) and Tigray from April 15 to May 10, 2018.”

Comment from reviewer #2

The introduction was written pretty well but still needs to show the consequences of inadequate health professionals and health extension worker motivation on the population health in general, such as death, a dropout from service, prolonged hospitalization, poor attitude to health facilities visits and soon. 

Authors response. Thank you for the insight included the consequences of low motivation to service on page4/5 line 72-75 reads as “Low health care workers motivation is the major contributor to the poor health service quality and can be associated with displeasure to clients, long waiting time, nonattendance, and unofficial fee charges and poor clinical outcome.”

Comment from reviewer #2

Method section 

1. Line page 10: check for typo-error such as … established through a “Scree” test and multiple… 

Authors response: Thank you for pointing this out. Word corrected as “scree test” on line179 of page11

2. Page 10, on lines 179 to 181, the authors mentioned that “the average job satisfaction was assessed by re-coding the 5-point Likert scale ranging from most satisfied with their job (1) to least satisfied (5) as a continuous variable.” Why did they need to consider scale one as most satisfied and scale five least? The revers will be more logical. Can you please explain this? Most studies and scholars use 5 as the most satisfied, while 1 is the least satisfied. 

Author Response: Thank you for pointing this out. The Author recognized as an error and corrected on page 12 of line 190-192 Reads as “Average job satisfaction was assessed by re-coding the 5-point Likert scale ranging from least satisfied with their job (1) to most satisfied (5) as a continuous variable.”

Comment from reviewer #2

Result section 

1. Table titles are not self-explanatory. Table/figure titles must be self-explanatory, which means every audience should be understood what the figure/table contains. 

Author response: thank you for pointing this out. Table & figure titles Corrected.

The corrected table and titles read as follows

Table 1. Motivation questions included in survey, source, and domain from April 15-May10,2018 (Line 116 on page6)

Table 2. Background characteristics of participants from four regions, (n=397), April 15-May10,2018 (Page13 line205)

Figure 1. Plot of responses to motivational survey items among participants from four regions of Ethiopia, April 15-May10,2018 (Page 14 of line213)

Table 3. Exploratory factor analysis results showing the factor loadings by individual items April 15- May 10, 2018 (line237 page16)

Table 4. Factors associated with overall motivation among participants form four regions April 15- May10, 2018 (line265 of page 18) 

Table 5. Association between motivation factors and demographic and structural factors among participants from four regions April 15- May10,2018. (Line 284 of page 20)

2. The number of the total participants is not consistently explained in the document. For example, the authors told us that only 397 participants were provided their responses. However, they explained 401 respondents in the table. Similarly, the authors should re-check that the sum of all participants is similar to the total respondents. E.g., in Table 2: region, the sum of 107, 106, 137, 50 is 400, which is not equal to the total participants explained by the title. Same table, data for age: the sum of 90, 232, 75 is …. Check other also.

Author response: thank you! The author corrected with the total respondents completed the survey n= 397 in the manuscript result section and tables as shown on line 205, table2 of page13.

3. Page 16, line 234/5… most demotivating factor mentioned by 29% … add the confidence interval. 

Authors response: thank you! Confidence interval on page 17 line247, read as “The most demotivating factor mentioned by 29% participants was workload (95% CI 34% - 39%).”

4. What are your criteria for selecting a reference group in the regression analysis? 

Author’s response:

The Authors have no reason to justify but the criteria for selecting the reference group were the random Alphabet order (Amhara).

Comment from reviewer #2

Discussion 

1. What is the reason or reference to say motivation among HEW is high? 

Author response: thank you. The Author is trying to explain that the respondents 61% of them were motivated to do their job. The Authors did not mention the specific group of professionals (HEW).

2. What is the possible reason health extension workers get motivated in the Amhara region compared to the SNNP? This might be good to scale up to other regions. 

Author response Thank you. From regression analysis the recognition and support motive among respondent were significantly lower in SNNPR & OROMIA as compared to Amhara region. This may explain that recognition and support in this region high. As studies show that recognition is a major predictor for motivation.

3. Based on your finding, what do you suggest/recommend for the concerned body? 

Authors response: thank you recommendation included, and these can be found in the manuscript page25 line 370-371

“Further studies are needed to explore the reason for variation in motivation across regions and cadre type.”

Supporting information:

#1Questionnnaire: All variables included in the analysis are explained fully in the text and figures, and we have sought to ensure that replication would be possible from these. Questionnaire used for data collection was done with ODK as in line 164-165. We will provide the ODK file which was developed in Amharic and English language up on reasonable request.

#2 Data: We will provide the data analyzed and interpreted DOI of the current manuscript up on reasonable request.

#3 Pretest: The survey was piloted out of 19 woreda health office staff in December 2017. No changes were made to the survey between piloting and the final survey as it was understood well by participants, assessed through debriefing interviews after survey completion as shown in line141-144.

---

## [Decision Letter · Decision Letter 1]

7 Feb 2022

PONE-D-21-09235R1Key factors influencing motivation among health extension workers and health care professionals in four regions of Ethiopia: a cross-sectional study.PLOS ONE

Dear Dr. Adillo,

Thank you for submitting your manuscript to PLOS ONE. After careful consideration, we feel that it has merit but does not fully meet PLOS ONE’s publication criteria as it currently stands. Therefore, we invite you to submit a revised version of the manuscript that addresses the points raised during the review process. Specifically there are a number of small grammar modifications and changes to the abstract that one reviewer has listed below..

Reviewer #1: Al comments have been addressed

Reviewer #2: All comments have been addressed

2. Is the manuscript technically sound, and do the data support the conclusions?

Reviewer #1: Yes

Reviewer #2: Yes

3. Has the statistical analysis been performed appropriately and rigorously? 

Reviewer #1: Yes

Reviewer #2: Yes

4. Have the authors made all data underlying the findings in their manuscript fully available?

Reviewer #1: Yes

Reviewer #2: Yes

5. Is the manuscript presented in an intelligible fashion and written in standard English?

Reviewer #1: Yes

Reviewer #2: Yes

6. Review Comments to the Author

Reviewer #1: (No Response)

Reviewer #2: Key factors influencing motivation among health extension workers and health care professionals in four regions of Ethiopia: a cross-sectional study

I am excited to get the opportunity to review the manuscript mentioned above. In general, the manuscript is a well-stated and essential area for the developing country. However, before accepting publication, the authors should consider the following issues.

Abstract

Include the objective of the study in the abstract section. Make sure to use the same abstract in the system and main manuscript. The abstract in the main manuscript has no objectives, while the abstract in the system contain it.

What does non-patient-facing health system staff mean? Who are they?

Please give space between a statement and citation throughout the manuscript.

Line 65 -66, what was the finding of the study? Why did the author explain these studies?

Even though the sample size is smaller, it includes four regions (Amhara, Oromo, SNNPR, and Tigray), which is good and improves the representativeness of the finding. But, what is your justification for including these? Why not Harari, Gambella…

Table 5: Association between motivation factors and 271 demographic and structural factors

Use consistent decimal numbers throughout the document. Example region: (-0.21, 0.006) 0.064

Discussion session

Page 22 and line 301-302: add the figure with confidence interval.

Page 22 and line 303: delete one “factor”.

Page 22 and lines 303-314: How do you think your finding is different from the previous one? Consider this comment for all results?

What is the limitation of your study?

7. PLOS authors have the option to publish the peer review history of their article (what does this mean?). If published, this will include your full peer review and any attached files.

Reviewer #1: No

Reviewer #2: **Yes: **Cheru Tesema Leshargie

---

## [Author Response · Author response to Decision Letter 1]

16 Feb 2022

Feb 16, 2022

Manuscript PONE-D-21-09235 

Response to reviewers 

Dear Editor in Chief

Thank you so much for the opportunity you have given us to re-submit a revised manuscript entitled “Key factors influencing motivation among health extension workers and health care professionals in four regions of Ethiopia: a cross sectional study.” for consideration for publication in PLOS ONE.

We appreciate the time and effort you and the reviewers dedicated to providing feedback to our manuscript and are grateful for the insightful comments on and valuable improvements made to our paper. 

We have incorporated most of the suggestions made by the reviewer. Those changes are highlighted with yellow color within the revised manuscript with track changes. Please see below a blue colored point-by-point response to the reviewers’ comments and concerns. Please note that all page numbers and line numbers refer to the revised manuscript.

Thank you so much once again! 

Reviewer #1 no comment 

Reviewer# 2: Key factors influencing motivation among health extension workers and health care professionals in four regions of Ethiopia: a cross-sectional study.

I am excited to get the opportunity to review the manuscript mentioned above. In general, the manuscript is a well-stated and essential area for the developing country. However, before accepting publication, the authors should consider the following issues.

Abstract: 

Include the objective of the study in the abstract section. Make sure to use the same abstract in the system and main manuscript. The abstract in the main manuscript has no objectives, while the abstract in the system contains it.

Authors Response: thank you for pointing out this. We have included the objective on page 2 line 26-27 and it reads as ’Objective this study sought to determine the extent and variation of health professionals’ motivation alongside factors associated with motivation.’ 

What does non-patient-facing health system staff mean? Who are they?

Authors response: thank you for asking, non-patient facing health system staffs are professionals working in different structures of a health system. It is stated on the abstract on page 2 line 32-33 as ’non-patient facing health system staff representing case team leaders, facility and district heads, directors, and officers (n=81).’ 

Please give space between a statement and citation throughout the manuscript.

Authors response: Thank you for the comment, corrected accordingly as shown on line 50, 53, 55, 65, 72, 106, 109, 188, 322, 325, 332 and 343. 

Line 65 -66

What was the finding of the study? Why did the author explain these studies?

Authors response: Thank you, the study assessed motivation and retention of health care workers and indicated importance of financial incentives on health workers motivation. In our study, there are different factors associated with motivation including financial incentives and revealed that financial and managerial support were significantly associated with health professionals job satisfaction, region, leave days and job title as specified on the result section on page 22 line 300 -310. These also further explained on the discussion section on page 24 on line 341 - 343 as ‘Recognition and support were significantly higher among health care providers than among HEWs. Having more recognition (financial and managerial) support was a positive predictor for job satisfaction and motivation.’

Even though the sample size is smaller, it includes four regions (Amhara, Oromo, SNNPR, and Tigray), which is good and improves the representativeness of the finding. But, what is your justification for including these? Why not Harari, Gambella.

Authors response: we believed that the four regions are home to more than 81% of the population in Ethiopia and accommodates most health facilities and health workers. These regions are highly diverse population and have different geographic characteristics improving representativeness as stated on page 6, line number 100 - 102.

Table 5: 

Association between motivation factors and 271 demographic and structural factors

Use consistent decimal numbers throughout the document. Example region: (-0.21, 0.006) 0.064

Authors response: Thank you for pointing this out, the decimal numbers corrected accordingly throughout the document. On page number 13 table 2 (region _Amhara %), on page number 17-line number 249, on page number 18 Table 4 perceived gross salary fair (p value), on page number 19 in line number 270, on page number 20-21 table 5 gender factor 2 (coef.), region factor 2 (coef.), factor 2 region (95%CI), age>30 factor1 (95%CI), work experience factor2 (coef.), factor3 job title (95%CI), on page number 22 in line number 299. 

Discussion session

Page 22 and line 301-302: add the figure with confidence interval.

Authors response: thank you for pointing out, we have included the interval accordingly it reads as ‘the mean score of recognition and support was 1.77 (95% CI 1.72,1.83; P=0.001)’on page 22 line 299.

Page 22 and line 303: delete one “factor”.

Authors response: thank you deleted accordingly.

Page 22 and lines 303 - 314: How do you think your finding is different from the previous one? Consider this comment for all results?

Authors response: line number 303-314 our study finding “motivation varied from region to region’’ is different in its scope that the study was conducted in four big regions and regional variation of motivation was studied at the same time. In addition, the health care workers motivation and health extension workers motivation were compared, and this study is different in computing multiple characteristics with the overall motivation and three motivation factors.

The three factors also indicated variation with associated factors (region, age, leave days, job satisfaction, perceived gross salary, workload…) as on page number 17-line number 249-257, on page number 19-line number 271-280, and on page number 22-line number 287 – 295. 

This has been addressed in the manuscript discussion section on page 24 and line number 345-349 stated as ‘A strength of this study was that data was collected from four (Amhara, Oromia, SNNPR, Tigray) major regions of Ethiopia where most of the country’s population live and the regional variation of motivation was studied at the same time, and the health care workers and health extension workers motivation were compared, in addition this study addressed computing multiple associated factors with the overall motivation and three motivation factors identified during factor analysis.’

What is the limitation of your study?

Authors response: thank you for this question, we have already stated a number of limitations in the manuscript on page24 & 25, line number 348-361:

 ‘Motivation was self-reported and may therefore be subject to acceptability biases in the face-to-face interviews conducted. Knowledge that interviews were conducted by interviewers from a public university and evaluating an IHI programme may have biased responses. The skilled care provider sample was too small to conduct a subgroup analysis among specific groups of interest such as doctors or midwives. Region-level analyses may have been under powered to variation in motivation, particularly in the presence of heterogeneity in motivation within regions, for example by cadre, or other unobserved variables. The sampling strategy was not optimized to account for or detect potentially important heterogeneity between regions, urban and rural areas, or QI programme areas, which may further reduce power to detect across-region variation. Qualitative work could have been conducted alongside quantitative data collection to understand why variation exists in motivation between regions and personal characteristics. Finally, we were unable to assess the link between health worker motivation and the quality of care provided which remains under-researched.’

---

## [Decision Letter · Decision Letter 2]

21 Apr 2022

PONE-D-21-09235R2Key factors influencing motivation among health extension workers and health care professionals in four regions of Ethiopia: a cross-sectional study.PLOS ONE

Dear Dr. Adillo,

Thank you for submitting your manuscript to PLOS ONE. After careful consideration, we feel that it has merit but does not fully meet PLOS ONE’s publication criteria as it currently stands. Therefore, we invite you to submit a revised version of the manuscript that addresses the points raised during the review process.

Please address the comment of the reviewer in the Discussion section (lines 315-316) by providing a reference  in a revised manuscript.

We look forward to receiving your revised manuscript.

Kind regards,

Colin Johnson, Ph.D.

Academic Editor

PLOS ONE

Journal Requirements:

Reviewers' comments:

Reviewer's Responses to Questions

**Comments to the Author**

1. If the authors have adequately addressed your comments raised in a previous round of review and you feel that this manuscript is now acceptable for publication, you may indicate that here to bypass the “Comments to the Author” section, enter your conflict of interest statement in the “Confidential to Editor” section, and submit your "Accept" recommendation.

Reviewer #2: All comments have been addressed

2. Is the manuscript technically sound, and do the data support the conclusions?

Reviewer #2: Yes

3. Has the statistical analysis been performed appropriately and rigorously? 

Reviewer #2: Yes

4. Have the authors made all data underlying the findings in their manuscript fully available?

Reviewer #2: Yes

5. Is the manuscript presented in an intelligible fashion and written in standard English?

Reviewer #2: Yes

6. Review Comments to the Author

Reviewer #2: All required previous questions have been answered and that all responses meet formatting specifications. I pointed some additional issues in the upladed document. The authors need to see the documment to address further question, which hopefully improve the quality of their work. By doing this, authors may contribute scientific documents to the readers.

7. PLOS authors have the option to publish the peer review history of their article (what does this mean?). If published, this will include your full peer review and any attached files.

Reviewer #2: **Yes: **Cheru Tesema Leshargie

---

## [Author Response · Author response to Decision Letter 2]

28 May 2022

Dear Editor in Chief

Thank you so much for the opportunity you have given us to re-submit a revised manuscript entitled “Key factors influencing motivation among health extension workers and health care professionals in four regions of Ethiopia: a cross sectional study.” for consideration for publication in PLOS ONE.

We appreciate the time and effort you and the reviewers dedicated to providing feedback to our manuscript and are grateful for the insightful comments and valuable improvements made to our paper. 

We have incorporated most of the suggestions made by the reviewer. Those changes are highlighted with yellow color within the revised manuscript with track changes. Please see below a blue colored point-by-point response to the reviewers’ comments and concerns. Please note that all page numbers and line numbers refer to the revised manuscript.

Thank you so much once again! 

Reviewer#1: No comment 

Reviewer# 2:

The abstract is not structured well. the authors need to restructure it clearly separating the important sections.

Authors response: Thank you. Yes, we separated all the important components of the abstract Background, Objectives, Methods, Results, and conclusion section without violating the PLOS ONE manuscript body submission guideline. The statement explained as 

“Background: Although Ethiopia has improved access to health care in recent years, quality of care remains low. Health worker motivation is an important determinant of performance and affects quality of care. Low health care workers motivation can be associated with poor health care quality and client experience, non-attendance, and poor clinical outcome. Objective his study sought to determine the extent and variation of health professionals’ motivation alongside factors associated with motivation. 

Methods: We conducted a facility based cross-sectional study among health extension workers (HEWs) and health care professionals in four regions: Amhara, Oromia, South nations, and nationalities people’s region (SNNPR) and Tigray from April 15 to May 10, 2018. We sampled 401 health system workers: skilled providers including nurses and midwives (n=110), HEWs (n=210); and non-patient facing health system staff representing case team leaders, facility and district heads, directors, and officers (n=81). Participants completed a 30-item Likert scale ranking tool which asked questions across 17 domains. We used exploratory factor analysis to explore latent motivation constructs. 

Results: Of the 397 responses with complete data, 61% (95% CI 56%-66%) self-reported motivation as “very good” or “excellent”. Significant variation in motivation was seen across regions with SNNPR scoring significantly lower on a five-point Likert scale by 0.35 points (P=0.003). The exploratory factor analysis identified a three-factors: personal and altruistic goals; pride and personal satisfaction; and recognition and support. The personal and altruistic goals factor varied across regions with Oromia and SNNPR being significantly lower by 0.13 (P=0.018) and 0.12 (P=0.039) Likert points respectively. The pride and personal satisfaction factor were higher among those aged >=30 years by 0.14 Likert scale points (P=0.045) relative to those aged between 19-24years. 

Conclusions: Overall, motivation was high among participants but varied across region, cadre, and age. Workload, leave, and job satisfaction were associated with motivation.” 

Reviewer#2

The method section needs to be restructured. Some important sections were still not included. Try to consider all necessary components.

Authors response: thank you, we included the necessary components “Study design” and moved the sentence under Materials and Methods sub section (study design) as stated in line 104-107 as “We conducted a facility based cross-sectional study among health extension workers (HEWs) and health care professionals in four regions: Amhara, Oromia, South nations, and nationalities people’s region (SNNPR) and Tigray from April 15 to May 10, 2018. “

In addition, on sub section (sampling and data collection) in line 131-133 reads as “We sampled 401 health system workers: skilled providers including nurses and midwives (n=110), HEWs (n=210); and non-patient facing health system staff representing case team leaders, facility and district heads, directors, and officers (n=81).”

Reviewer#2

Line136: Why purposive? How do you analyze data obtained from the two randomly and nonrandom?

Authors response: thank you. Purposive sampling was done for two Woreda which had simultaneous qualitative work ongoing in the broader QI evaluation (Page 144).

There was no evidence (before or after) analysis that key dimensions or results differed between randomly and non-randomly chosen woreda, so these data are pooled together. This analysis strategy was also taken in two previously published papers using this dataset:

Quaife, M., Estafinos, A. S., Keraga, D. W., Lohmann, J., Hill, Z., Kiflie, A., ... & Schellenberg, J. (2021). Changes in health worker knowledge and motivation in the context of a quality improvement programme in Ethiopia. Health policy and planning, 36(10), 1508-1520.

Lamba, S., Arora, N., Keraga, D. W., Kiflie, A., Jembere, B. M., Berhanu, D., ... & Quaife, M. (2021). Stated job preferences of three health worker cadres in Ethiopia: a discrete choice experiment. Health Policy and Planning, 36(9), 1418-1427.

Reviewer#2

Line 155, Why 50? Some regions contain larger number of the population Line 162

Authors response: 

As stated on in line 163-166 “A target sample size of 50 respondents per region was chosen, based on the primary research question of assessing changes in motivation as measured by Likert scale questions, in line with a rule of thumb in exploratory factor analysis that 50 participants per cluster is a reasonable sample size to detect differences across clusters (21)”. Weighting the number of respondents by region size would not have led to more generalizable inference.

Reviewer #2 Result section line 202, 203 Add frequency 

Authors response:

Thank you included the frequency as on line 210-211. Reads as “Two hundred five (51%) of respondents had greater than 4 years of work experience, and (208, 52.39%) of the respondents were HEWs shown in Table 2.”

Reviewer #2

Page13, 18 and 21 in Table 2,4 & 5: (job title Other) who are this try to explain or mention all 

Authors response: Thank you. We included all as Other (facility and district head, directors, and officers), and Leaders as case team leader, on page 13, 18, 21 in table 2, 4 and 5.

Reviewer #2

Discussion: How high is it? Where is the figure? What was your reference to conclude finding is high? How much is the variation among region and cadre?

Authors response: 

Thank you for pointing out this. We corrected the wording as per the comment. 

Our study has revealed the extent of motivation indicating 61% (majority of the participants were motivated to do their job). We stated as “Our study results indicated that more than 60% of HEW and health workers were motivated to do their job. Key factors identified to influence motivation were region, age, job title, work experience, job satisfaction, and leave days.” 

We also already cited references from previous studies to compare with our study. Though, the studies were conducted among health professionals working in hospitals. 

The study has also indicated the regional variation in general was under powered as already mentioned as a limitation in line 367-373.

---

## [Editor Report · Decision Letter 3]

31 May 2022

PONE-D-21-09235R3Key factors influencing motivation among health extension workers and health care professionals in four regions of Ethiopia: a cross-sectional study.PLOS ONE

Dear Dr. Adillo,

Thank you for submitting your manuscript to PLOS ONE. After review, Reviewer 2 has a few text suggestions to improve the manuscript. Please examine the suggestions and make changes where you believe it improves the manuscript.

We look forward to receiving your revised manuscript.

Kind regards,

Colin Johnson, Ph.D.

Academic Editor

PLOS ONE
---

## [Author Response · Author response to Decision Letter 3]

29 Jun 2022

Dear Editor in Chief

Thank you so much for the opportunity you have given us to re-submit a revised manuscript entitled “Key factors influencing motivation among health extension workers and health care professionals in four regions of Ethiopia: a cross sectional study.” for consideration for publication in PLOS ONE.

We appreciate the time and effort you and the reviewers dedicated to providing feedback to our manuscript and are grateful for the insightful comments and valuable improvements made to our paper. 

We have incorporated most of the suggestions made by the reviewer. Those changes are highlighted with yellow color within the revised manuscript with track changes. Please see below a blue colored point-by-point response to the reviewers’ comments and concerns. Please note that all page numbers and line numbers refer to the revised manuscript.

Thank you so much once again! 

Reviewer#1: No comment 

Reviewer# 2:

The abstract is not structured well. the authors need to restructure it clearly separating the important sections.

Authors response: Thank you. Yes, we separated all the important components of the abstract Background, Objectives, Methods, Results, and conclusion section without violating the PLOS ONE manuscript body submission guideline. The statement explained as 

“Background: Although Ethiopia has improved access to health care in recent years, quality of care remains low. Health worker motivation is an important determinant of performance and affects quality of care. Low health care workers motivation can be associated with poor health care quality and client experience, non-attendance, and poor clinical outcome. Objective this study sought to determine the extent and variation of health professionals’ motivation alongside factors associated with motivation. 

Methods: We conducted a facility based cross-sectional study among health extension workers (HEWs) and health care professionals in four regions: Amhara, Oromia, South nations, and nationalities people’s region (SNNPR) and Tigray from April 15 to May 10, 2018. We sampled 401 health system workers: skilled providers including nurses and midwives (n=110), HEWs (n=210); and non-patient facing health system staff representing case team leaders, facility and district heads, directors, and officers (n=81). Participants completed a 30-item Likert scale ranking tool which asked questions across 17 domains. We used exploratory factor analysis to explore latent motivation constructs. 

Results: Of the 397 responses with complete data, 61% (95% CI 56%-66%) self-reported motivation as “very good” or “excellent”. Significant variation in motivation was seen across regions with SNNPR scoring significantly lower on a five-point Likert scale by 0.35 points (P=0.003). The exploratory factor analysis identified a three-factors: personal and altruistic goals; pride and personal satisfaction; and recognition and support. The personal and altruistic goals factor varied across regions with Oromia and SNNPR being significantly lower by 0.13 (P=0.018) and 0.12 (P=0.039) Likert points respectively. The pride and personal satisfaction factor were higher among those aged >=30 years by 0.14 Likert scale points (P=0.045) relative to those aged between 19-24years. 

Conclusions: Overall, motivation was high among participants but varied across region, cadre, and age. Workload, leave, and job satisfaction were associated with motivation.” 

Reviewer#2

The method section needs to be restructured. Some important sections were still not included. Try to consider all necessary components.

Authors response: thank you, we included the necessary components “Study design” and moved the sentence under Materials and Methods sub section (study design) as stated in line 104-107 as “We conducted a facility based cross-sectional study among health extension workers (HEWs) and health care professionals in four regions: Amhara, Oromia, South nations, and nationalities people’s region (SNNPR) and Tigray from April 15 to May 10, 2018. “

In addition, on sub section (sampling and data collection) in line 131-133 reads as “We sampled 401 health system workers: skilled providers including nurses and midwives (n=110), HEWs (n=210); and non-patient facing health system staff representing case team leaders, facility and district heads, directors, and officers (n=81).”

Reviewer#2

Line136: Why purposive? How do you analyze data obtained from the two randomly and nonrandom?

Authors response: thank you. Purposive sampling was done for two Woreda which had simultaneous qualitative work ongoing in the broader QI evaluation (Page 144).

There was no evidence (before or after) analysis that key dimensions or results differed between randomly and non-randomly chosen woreda, so these data are pooled together. This analysis strategy was also taken in two previously published papers using this dataset:

Quaife, M., Estafinos, A. S., Keraga, D. W., Lohmann, J., Hill, Z., Kiflie, A., ... & Schellenberg, J. (2021). Changes in health worker knowledge and motivation in the context of a quality improvement programme in Ethiopia. Health policy and planning, 36(10), 1508-1520.

Lamba, S., Arora, N., Keraga, D. W., Kiflie, A., Jembere, B. M., Berhanu, D., ... & Quaife, M. (2021). Stated job preferences of three health worker cadres in Ethiopia: a discrete choice experiment. Health Policy and Planning, 36(9), 1418-1427.

Reviewer#2

Line 155, Why 50? Some regions contain larger number of the population Line 162

Authors response: 

As stated on in line 163-166 “A target sample size of 50 respondents per region was chosen, based on the primary research question of assessing changes in motivation as measured by Likert scale questions, in line with a rule of thumb in exploratory factor analysis that 50 participants per cluster is a reasonable sample size to detect differences across clusters (21)”. Weighting the number of respondents by region size would not have led to more generalizable inference.

Reviewer #2 Result section line 202, 203 Add frequency 

Authors response:

Thank you included the frequency as on line 210-211. Reads as “Two hundred five (51%) of respondents had greater than 4 years of work experience, and (208, 52.39%) of the respondents were HEWs shown in Table 2.”

Reviewer #2

Page13, 18 and 21 in Table 2,4 & 5: (job title Other) who are this try to explain or mention all 

Authors response: Thank you. We included all as Other (facility and district head, directors, and officers), and Leaders as case team leader, on page 13, 18, 21 in table 2, 4 and 5.

Reviewer #2

Discussion: How high is it? Where is the figure? What was your reference to conclude finding is high? How much is the variation among region and cadre?

Authors response: 

Thank you for pointing out this. We corrected the wording as per the comment. 

Our study has revealed the extent of motivation indicating 61% (majority of the participants were motivated to do their job). We stated as “Our study results indicated that more than 60% of HEW and health workers were motivated to do their job. Key factors identified to influence motivation were region, age, job title, work experience, job satisfaction, and leave days.” 

We also already cited references from previous studies to compare with our study. Though, the studies were conducted among health professionals working in hospitals. 

The study has also indicated the regional variation in general was under powered as already mentioned as a limitation in line 367-373. 

Reviewers comment 

The number of the total participants is not consistently explained in the document. For example, the authors told us that only 397 participants were provided their responses. However, they explained 401 respondents in the table. Similarly, the authors should re-check that the sum of all participants is similar to the total respondents. E.g., in Table 2: region, the sum of 107, 106, 137, 50 is 400, which is not equal to the total participants explained by the title. Same table, data for age: the sum of 90, 232, 75 is …. Check other also. 

Authors Response 

Thank you! 

We have checked and corrected the number of participants during the 1st and 2nd review period (earlier versions). 

Currently, we also included a sentence to further clarify the surveyed participants and response rate in line 202 as “Of 401 people surveyed, 397 responded complete giving a response rate of 99%.”

---

## [Editor Report · Decision Letter 4]

22 Jul 2022

Key factors influencing motivation among health extension workers and health care professionals in four regions of Ethiopia: a cross-sectional study.

PONE-D-21-09235R4

Dear Dr. Adillo,

We’re pleased to inform you that your manuscript has been judged scientifically suitable for publication and will be formally accepted for publication once it meets all outstanding technical requirements.

Kind regards,

Colin Johnson, Ph.D.

Academic Editor

PLOS ONE
---

## [Editor Report · Acceptance letter]

8 Sep 2022

PONE-D-21-09235R4 

Key factors influencing motivation among health extension workers and health care professionals in four regions of Ethiopia: a cross-sectional study. 

Dear Dr. Quaife:

I'm pleased to inform you that your manuscript has been deemed suitable for publication in PLOS ONE. Congratulations! Your manuscript is now with our production department. 

Kind regards, 

on behalf of

Dr. Colin Johnson 

Academic Editor

PLOS ONE